# The Role of the Adrenal–Gut–Brain Axis on Comorbid Depressive Disorder Development in Diabetes

**DOI:** 10.3390/biom13101504

**Published:** 2023-10-10

**Authors:** Thalita Mázala-de-Oliveira, Bruna Teixeira Silva, Paula Campello-Costa, Vinicius Frias Carvalho

**Affiliations:** 1Laboratório de Inflamação, Instituto Oswaldo Cruz, Fundação Oswaldo Cruz, Rio de Janeiro 21040-360, Brazil; tha.mazala@hotmail.com (T.M.-d.-O.); bruna_teixeira@id.uff.br (B.T.S.); 2Programa de Pós-Graduação em Neurociências, Instituto de Biologia, Universidade Federal Fluminense, Niterói 24210-201, Brazil; paulacampello@id.uff.br; 3Laboratório de Inflamação, Instituto Nacional de Ciência e Tecnologia em Neuroimunomodulação—INCT-NIM, Instituto Oswaldo Cruz, Fundação Oswaldo Cruz, Rio de Janeiro 21040-360, Brazil

**Keywords:** treatment-resistant depression, depression, diabetes, glucocorticoids, gut microbiota, TLR4

## Abstract

Diabetic patients are more affected by depression than non-diabetics, and this is related to greater treatment resistance and associated with poorer outcomes. This increase in the prevalence of depression in diabetics is also related to hyperglycemia and hypercortisolism. In diabetics, the hyperactivity of the HPA axis occurs in parallel to gut dysbiosis, weakness of the intestinal permeability barrier, and high bacterial-product translocation into the bloodstream. Diabetes also induces an increase in the permeability of the blood–brain barrier (BBB) and Toll-like receptor 4 (TLR4) expression in the hippocampus. Furthermore, lipopolysaccharide (LPS)-induced depression behaviors and neuroinflammation are exacerbated in diabetic mice. In this context, we propose here that hypercortisolism, in association with gut dysbiosis, leads to an exacerbation of hippocampal neuroinflammation, glutamatergic transmission, and neuronal apoptosis, leading to the development and aggravation of depression and to resistance to treatment of this mood disorder in diabetic patients.

## 1. Introduction

Diabetes mellitus is a group of metabolic diseases characterized by disturbances in the homeostasis of carbohydrate metabolism that result in hyperglycemia. Currently, diabetes is one of the most serious and common chronic diseases worldwide. Patients with uncontrolled diabetes develop several disabling complications that reduce their life expectancy, and which can even be fatal. In 2021, the global prevalence of diabetes reached pandemic proportions with 537 million people with the disease worldwide, accompanied by a global health expenditure of US $966 billion [1]. Furthermore, future projections suggest that by 2045 the number of people with diabetes will increase by 46% [1], and the estimated health expenditure for the care of this disease that will exceed USD one trillion.

Diabetic patients are more susceptible to developing mood disorders, including depression, than non-diabetic individuals. Diabetic patients who have mood disorders have a high risk of mortality from micro- and macrovascular complications when compared to diabetic patients who do not develop these comorbidities [2,3,4]. In addition, diabetes-related mood disorders culminate in a large economic burden for patient care and are associated with more treatment-resistant depression [5,6]. Therefore, knowledge of the mechanisms underlying the establishment of mood disorders in diabetics is fundamental for the early diagnosis and/or treatment of patients. The pathophysiology of mood disorders in diabetics is complex and multifactorial, and includes factors inherent to diabetes itself, such as hyperglycemia and diabetes-related microvascular dysfunction. Furthermore, changes in hippocampal homeostasis, including neuroinflammation, oxidative stress, imbalance in neurotransmitter levels, and decrease in brain-derived neurotrophic factor (BDNF), are key to the evolution of mood disorders in diabetic patients [7,8,9,10].

It is well known that there is a bidirectional communication between the neuroendocrine system, including the hypothalamus–pituitary–adrenal (HPA) axis, which is the central stress-response system, and gut microbiota [11,12,13]. The HPA axis is a neuroendocrine system responsible for the stress response triggered by both internal and external stimuli. Activation of the HPA axis triggers neurons in the paraventricular nucleus (PVN) of the hypothalamus to release corticotropin-releasing hormone (CRH), which subsequently induces secretion of adrenocorticotropic hormone (ACTH) by the anterior pituitary and, finally, glucocorticoids from the adrenal cortex. Due the metabolic effects of chronic exposure to high glucocorticoids levels, the HPA axis needs to be finely regulated. Therefore, rising levels of glucocorticoids activate their receptors (GRs) in the hypothalamus and pituitary, inhibiting further release of CRH and ACTH, in a classic endocrine negative feedback loop, which enables the HPA axis to return to a physiological state following acute activation [14].

The human gut microbiota is composed of up to 100 trillion complex microorganisms, such as commensal, symbiotic, and pathogenic bacteria, as well as archaea, fungi, and viruses that colonize the intestine [15]. Exogenous chronic treatment with glucocorticoids, the major effector hormone produced by the HPA axis, shifted the composition of gut microbiota and the profile of fecal metabolites in rats [11], reducing the fecal production of short-chain fatty acids (SCFAs). In addition, chronic stress promotes changes in the gut microbiota profile and leaks in the epithelial–intestinal barrier [16]. In addition, maternal separation, a powerful stressor in early life, increases plasma corticosterone and leads to changes in the microbiota and systemic immune response, revealing alterations in the intestine–microbiota–brain axis [17]. On the other hand, the gut microbiome composition can regulate the activity of the HPA axis. The lower diversity and higher relative abundance of pathogenic bacteria induced by maternal precarity were positively correlated with hyperactivity of the HPA axis [18]. Furthermore, germ-free mice, those with no commensal microbiota, exhibited higher systemic ACTH and corticosterone levels after acute stress response than specific pathogen-free (SPF) mice [19,20] and pre-treatment with probiotic *Lactobacillus farciminis* inhibited acute psychological stress-induced HPA axis hyperactivity in rats [19,20].

Since patients with both diabetes and depression showed a hyperactivity of the HPA axis accompanied by gut dysbiosis [21,22,23,24], a term commonly used to describe a bloom of pathobionts, loss of commensals, and/or loss of diversity [25], we hypothesized that disturbance in the adrenal–gut–brain axis is a crucial factor in diabetes-induced comorbid depressive disorder development and treatment-resistant depression.

## 2. Diabetes, the HPA Axis, and Microbiota

It is well known that diabetic patients present a hyperactivity of the HPA axis, evidenced by an increase in the circulating levels of ACTH and cortisol as well as elevated urinary free cortisol levels [26,27,28]. In diabetic patients, the hyperactivity of the HPA axis was related to both a failure in the negative feedback of the axis, attested by the dexamethasone suppression test [28], and a higher reactivity of adrenal to CRH stimulation [29]. We and others have shown that diabetic animals exhibit high levels of ACTH and corticosterone in the circulation [30,31,32,33], and an impairment in the negative feedback of the HPA axis [34]. We previously demonstrated that the exacerbation of glucocorticoid production by diabetic animals was related to overexpression of ACTH receptor (MC2R) and steroidogenic enzymes, including steroidogenic acute regulatory protein (StAR) and 11β-Hydroxysteroid dehydrogenase 1 (11β-HSD1), in the adrenal gland [33,35], while the failure in the negative feedback of the HPA axis was associated with a downregulation of GRs and mineralocorticoid receptors (MRs) in the pituitary. In addition, we showed that the reduction in the GR expression in the pituitary gland of diabetic rats occurred 24 h after the onset of hyperglycemia, suggesting that the lack of glycemic control in diabetics may be involved with the downregulation of GR in the pituitary gland [33,36].

Furthermore, we noted that diabetic animals showed a reduction in the anti-inflammatory receptor peroxisome proliferator-activated receptor γ (PPAR-γ) expression and an increase in the pro-inflammatory receptor angiotensin II type 1 receptor (AT1R) expression in the adrenal gland. In addition, the blocked of Ang-II/AT1R axis, which is a pro-inflammatory pathway, with captopril or olmesartan inhibited the exacerbation of corticosterone production by adrenal glands of diabetic mice through a mechanism dependent of a reduction in the expression of Ang-II and ACTH receptors, AT1R and MC2R, respectively, and steroidogenic enzymes StAR and 11β-HSD1. On the other hand, the activation of anti-inflammatory receptors PPAR-γ and AT2R with rosiglitazone and CGP42112A, respectively, inhibited the hypercortisolism in diabetic mice. Rosiglitazone induced a reduction in the circulating corticosterone levels which was related to a decrease in the systemic ACTH levels and in the expression of MC2R in the adrenal gland. Furthermore, the inhibitory effect of rosiglitazone on the hyperactivity of the HPA axis observed in diabetic rats was related to an upregulation of PI3K expression in the pituitary and adrenal glands. However, the exact mechanisms by which CGP42112A reduced the exacerbation of corticosterone production by adrenal glands of diabetic mice still remain elusive [35,36]. As observed in the adrenal gland, diabetic patients and animals showed a pro-inflammatory profile in their circulation [37,38]. In diabetics, the composition of the gut microbiota is altered and has been appointed as a driver of chronic low-grade systemic metabolic inflammation [39,40].

Patients with diabetes, both type 1 and type 2, presented a gut microbial dysbiosis [39,41,42]. In general, diabetic patients showed an increase in the ratio of Bacteroidetes/Firmicutes [41,43]. In addition, in all diabetic groups, a significant increase in the abundance of Gram-negative and potentially opportunistic pathogenic bacteria, such as *Pseudomonas* and *Prevotella*, in contrast to a reduction in the commensal bacteria, including *Turicibacter*, *Terrisporobacter*, and *Clostridium* was observed [44]. Patients with type 1 diabetes presented a clear depletion of species like *Prevotella copri* and *Bifidobacterium longum*, probiotic bacteria, and enrichment of families like *Ruminococcaceae*, *Clostridiaceae*, *Clostridiales*, and *Oscillibacter*, bacteria associated with infection and inflammation [45]. Furthermore, type 1 and type 2 diabetic patients presented a decrease in the abundance of SCFA-producing bacteria [46].

SCFAs are important to the maintenance of epithelial–intestinal barrier permeability, reducing the translocation of bacteria and their products from the gut to the bloodstream [47]. The reduction in the abundance of SCFA-producing bacteria in the gut microbiota is possibly involved with the breakdown of the intestinal epithelial barrier observed in diabetic patients, attested by the measure of circulating levels of lipopolysaccharide (LPS) and/or intestinal fatty acid binding protein [48,49]. Furthermore, LPS, a component of the cell wall of Gram-negative bacteria and potentially opportunistic pathogenic bacteria found in the gut microbiota of diabetic patients, enhanced intestinal permeability in vitro and in vivo [50,51]. Additionally, the levels of LPS, which has been proposed as a source for chronic low-grade systemic metabolic inflammation [52,53], LPS-binding protein (LBP) [52], and/or bacterial DNA, including *Proteobacteria,* are increased in the circulation of diabetic patients [54].

The maintenance of homeostasis in the gut environment is important not only in slowing down diabetes development but it is also central in the control of its complications [55], suggesting that dysbiosis may have an important role in diabetes-induced comorbid depressive disorder development and treatment-resistant depression.

## 3. Depression, the HPA Axis, and Microbiota

Depression, also known as major depression or major depressive disorder (MDD), is a psychiatric disease that affects how a person feels, thinks, and handles daily activities, such as sleeping, eating, or working. Currently, depression is the leading cause of disability worldwide, and is associated with an increased risk of suicide [56,57]. Furthermore, about 30% of patients with depression fail to respond to conventional treatment, leading to treatment-resistant depression [58,59]. The pathophysiology of depression is associated with genetic and environment factors [60,61]. It also emerges as a comorbidity of chronic and systemic medical illnesses such as diabetes mellitus [62,63]. Patients with depression showed a reduction in the hippocampal volume [64,65,66], which is associated with a decrease in the size of pyramidal neurons, retraction of dendrites, reduction of neurogenesis in the hippocampal dentate gyrus, and a drop in the astrocyte counts [67,68,69]. In animal models of depression, treatment with antidepressants restored the hippocampal volume by restoring neurogenesis and cell differentiation [70,71].

At the molecular level, several alterations in the homeostasis of hippocampal function are involved in the development and/or aggravation of depression, including impairment in the biogenic amines and glutamate (Glu) signaling [72], decrease in the BDNF levels [73], imbalance in the neurogenesis, apoptosis, and autophagia [74], and increase in oxidative stress and neuroinflammation [75,76] (Figure 1).

Interestingly, depression is anticipated by chronic exposure to stress [77,78], suggesting that activation of the HPA axis may be involved in the development of depression that cannot be explained only by imbalance in the hippocampal neurotransmitter levels. In fact, around 50% of depressed patients and 80% of severely depressed patients showed hyperactivity of the HPA axis [79,80]. In addition, depressive patients presented adrenal hypertrophy and an increase in the circulating levels of both ACTH and cortisol [81,82,83,84]. Furthermore, repeated exposure to short-term stress or injection of glucocorticoids induced depressive-like behavior in mice [85]. Dexamethasone inhibited the proliferation of rodent neural stem cells, suggesting that glucocorticoids can impact neurogenesis, as observed in depressed patients [86]. Both chronic severe stress, which caused increased glucocorticoid levels, and direct glucocorticoid administration induced neural cell death, atrophy of neuronal processes, and decrease in the hippocampal neurogenesis [87,88,89]. Furthermore, glucocorticoid-receptor-impaired (GR-i) mice, a transgenic mouse model of reduced GR-induced negative feedback regulation of the HPA axis, exhibited changes in depressive-like behaviors, together with a reduction in cell proliferation and imbalanced levels of neuroplastic and epigenetic markers in the hippocampus. The treatment of GR-i mice with agomelatine, an antidepressant drug that acts as a melatonin receptor agonist and a serotonin (5-HT) 2C receptor antagonist, improved depressive-like behaviors and reversed the deficit in hippocampal cell proliferation in GR-i mice [90]. Altogether, these data suggest that hyperactivity of the HPA axis can be a trigger for depression development and worsening.

Additionally, patients with post-traumatic stress disorder (PTSD) showed a decrease in serum 5-HT levels and an increase in circulating cortisol levels [91,92]. This imbalance in 5-HT and cortisol levels in patients with PTSD was associated with a dysregulation in the immune response, attested by high levels of circulating pro-inflammatory cytokines, such as IL-1β, IL-6, and TNF-α [92,93,94,95,96], and low levels of anti-inflammatory cytokines, including IL-4 and IL-10 [97]. It is noteworthy that elevated serum IL-6 and TNF-α levels, as well as increased levels of salivary cortisol, are also associated with treatment-resistant depression [98,99,100]. Treatment-resistant depression is a subset of MDD, which affects approximately 30% of patients and is diagnosed when MDD patients show an inadequate response to at least two trials of antidepressant pharmacotherapy [101]. Treatment-resistance depression occurs when patients develop the same underlying neurobiology as the treatment-responsive illness; however, the pathophysiological alterations are more severe such that standard treatment is inadequate, and/or patients present progressive neurobiological changes or iatrogenic effects compared to treatment-sensitive patients treated with first-choice antidepressants [102].

A subgroup of PTSD patients with cirrhosis exhibited lower microbial diversity in the gut microbiota, as well as an increase in the levels of pathobiontic bacteria, including *Enterococcus* and *Escherichia-Shigella* genera, and a reduction in the levels of *Lachnospiraceae* and *Ruminococcaceae* families [103]. Although the enterochromaffin cells, with the cooperation of gut microbiota, are responsible for most of the 5-HT production by humans [104], there is no correlation between the drop in 5-HT and gut dysbiosis in these patients. In addition, rats that have be induced to a PTSD-like phenotype showed changes in the levels of phyla Firmicutes, Bacteroidetes, Cyanobacteria, and Proteobacteria and a significant reduction in the 5-HT concentrations in the cerebral cortex [105]; however, there is no evidence that the decrease in the 5-HT levels in the brain was caused by a lower production of this neurotransmitter by enterochromaffin cells. A possible explanation for the reduction in 5-HT levels in the brain of patients with PTSD is an alteration in the activity of monoamine oxidase A (MAO-A), which is the major enzyme responsible for the metabolism of monoamines. In fact, the 5-HT concentration in the hippocampi decreased in parallel to an increase in the MAO-A activity in rats that had be induced to a PTSD-like phenotype [106]. Recently, it was shown that the treatment with the probiotic *Bacillus coagulans* Unique IS-2^®^ reduced chronic stress-induced depression in rats by a mechanism related to an increase in BDNF and 5-HT levels and a decrease in TNF-α, IL-1β, and dopamine levels in the hippocampus and frontal cortex. Also, probiotic treatment restored systemic levels of L-tryptophan, L-kynurenine, kynurenic-acid, and 3-hydroxyanthranilic acid, villi/crypt ratio, goblet-cell count, Firmicutes to Bacteroides ratio, and levels of acetate, propionate, and butyrate in fecal samples of rats subjected to chronic stress, suggesting that the antidepressive-like effect of probiotics may be due to a remodeling of the intestine–microbiota–brain axis [107].

The use of two approaches to avoid glucocorticoid action, the removal of adrenal glands, or treatment with GR antagonist RU486 (mifepristone), increased hippocampal neurogenesis [108,109,110], strongly suggesting that hypercortisolism is important to the development and/or aggravation of depression. In fact, treatment with mifepristone improved neurocognitive impairment in bipolar depressed patients [111]. Nevertheless, in other studies, this treatment leds to improvement in psychotic but not in depressive symptoms in patients with psychotic major depression [112,113].

In addition to neuroendocrine imbalance, modifications in the gut microbiome have also been correlated with depression. For instance, several studies have shown an imbalance in the Firmicutes/Bacteroidetes ratio in patients with MDD [113,114,115,116]. Furthermore, the fecal microbiota of patients with MDD presented a significant reduction in the diversity of commensal bacteria genera, such as *Clostridium* [117,118], the vast majority of whose species are butyrate-producing bacteria [119], *Bifidobacterium* and *Lactobacillus* [120], and an increase in the Gram-negative and opportunistic pathogenic bacterium, such as *Prevotella* and *Klebsiella* [114], indicating that these patients exhibited gut dysbiosis. Furthermore, the use of ketamine, a N-methyl-D-aspartate (NDMA) antagonist with an antidepressant effect [121], also increased the number of butyrate-producing bacteria [122] in a chronic social defeat stress model of depression in mice.

Interestingly, germ-free mice exhibited depressive-like behaviors when receiving fecal microbiota transplantation derived from depressive patients [123]. In addition, chronic unpredictable mild stress mice treated with *Clostridium butyricum* improved depressive-like behavior, upregulating 5-HT and BDNF levels in the brain [124]. Another study showed that the administration of *C. butyricum* in association with antidepressants reduced 70% of depressive symptoms in patients with treatment-resistant depressive disorder [125]. This evidence strongly suggests that as well as HPA axis hyperactivity, gut dysbiosis promotes the development and/or worsening of the depressive condition.

## 4. Adrenal–Gut Axis Imbalance Is Central to Depressive Disorder Development and Aggravation in Diabetes

In diabetes, the development of depressive disorder is related to hyperglycemia and an increase in the circulating pro-inflammatory cytokines IL-1β, IL-6, and TNF-α, known as low-grade inflammation [126,127,128]. It is well known that alterations in hippocampal homeostasis, such as neuroinflammation, oxidative stress, and imbalance in neurotransmitter and BDNF levels, are crucial to the progression of mood disorders during diabetes. Indeed, type-2-diabetes-induced depression was accompanied by an increase in pro-inflammatory cytokines, including TNF-α, and a reduction in the BDNF content in the hippocampus. Interestingly, the levels of agmatine, an arginine metabolite involved in the regulation of insulin secretion and neuroprotection, were reduced in the hippocampus of type 2 diabetic rats. However, a chronic systemic treatment with agmatine improved depressive-like symptoms and inhibited neuroinflammatory markers [129].

The hyperactivity of the HPA axis and the consequent elevated systemic glucocorticoid levels is the potential second hit in the comorbid depression observed in diabetic patients [23,130,131]. Diabetes induced an increase in the levels of both corticosterone and its receptor in the hippocampus of rats [132,133]. Since the hippocampus is the major region of the central nervous system affected by depression, these data suggest that depressive disorder development and aggravation in diabetes may be related to hypercortisolism. In an in vitro model of diabetes-induced depression, using a hippocampal neurovascular unit (NVU) that contained hippocampal neurons, astrocytes, and microvascular endothelial cells from rats incubated in a hyperglycemic milieu, the activation of GR by corticosterone caused an impairment in the barrier function in clear association with neuronal apoptosis in the hippocampal NVU. This neuronal apoptosis in hippocampal NVU after the induction of diabetes-related depression in vitro was related to an activation of both GR and metabotropic glutamate receptor 2/3 (mGluR2/3) [4]. Interestingly, the treatment of *db/db* mice, a spontaneously murine model of type 2 diabetes, with metyrapone, which is a corticosterone synthesis inhibitor, reduced systemic corticosterone levels in parallel to a decrease in hippocampal levels of pro-inflammatory cytokines IL-1β and TNF-α and IBA1+ cells [134]. In addition, streptozotocin-induced diabetic mice showed a reduction in the astrocytic reactivity, attested by GFAP expression, in the hippocampus, which was sensitive to GR antagonist treatment [135]. Altogether, these data indicate that depressive disorder evoked by diabetes is related to a hypercortisolism-induced increase in the neuronal apoptosis and a reduction in astrocytic reactivity in the hippocampus (Figure 1).

In diabetic patients, hyperglycemia and low-grade inflammation are capable of disrupting the BBB, especially in the hippocampus region, and the epithelial–intestinal barrier [136,137,138,139]. Furthermore, glucocorticoid-induced brain damage is related to neuroinflammation. Although glucocorticoids suppress inflammation systemically, they can realize either pro- or anti-inflammatory actions in the brain, depending on the degree and duration of exposure, external factors preceding injury, injury characteristics, and the specific brain region [140,141]. For instance, if stress or glucocorticoid administration occurred prior to LPS injection directly into the hippocampus, it potentiated pro-neuroinflammatory effects of the LPS [142]. Chronic exposure to stress or glucocorticoid administration also potentiated LPS-induced neuroinflammation [143,144]. Glucocorticoids can also induce dysbiosis in parallel with a disruption in the epithelial–intestinal barrier, as observed in diabetics [11,135,145]. Altogether, the leak in both the epithelial–intestinal barrier and the BBB allows endotoxins, including LPS, present in the pathogenic bacteria that make up the intestinal microbiota of diabetics, to reach the hippocampus. In addition, the hippocampal neuronal apoptosis observed in diabetic mice was related to an increase in the expression of TLR4 [146] (Figure 1).

The biguanide metformin, which is a first-line drug for the treatment of type 2 diabetic patients, was effective in reducing depressive symptoms in non-diabetic and diabetic patients [147,148]. Interestingly, patients with type 2 diabetes treated with metformin showed alterations in the diversity of gut microbiota compared with those treated with a placebo, and the treatment significantly increased the concentrations of fecal SCFA, including propionate and butyrate, and plasma bile acids, such as primary, secondary, and unconjugated bile acids. The metformin-induced increase in these gut microbiota metabolites in type 2 diabetic patients was related to its antidiabetic effects [149]. The treatment with metformin also reduced the serum corticosterone levels in high-fat-diet-induced obese rats [150]. Furthermore, metformin improved LPS-induced depression in mice, through normalization of glutamatergic transmission in the hippocampal CA1 pyramidal neurons [151], suggesting that TLR4 activation in the hippocampus of diabetics by pathogenic products from the gut microbiota can be crucial for the aggravation of depression disorder. In addition, high glucose enhanced LPS-induced microglia activation in vitro, in clear association with an increase in TLR4 expression [152,153]. Interestingly, stimulation with LPS induced neuroinflammation, resulting in profound changes in the ultrastructure of NVU [154], as observed in diabetic mice [155,156], suggesting that TLR4 activation-induced neuroinflammation can increase the breakdown of BBB observed in diabetics. Moreover, diabetes increased LPS-induced depression behaviors and neuroinflammation in mice [157].

Finally, there is a close association between HPA axis disruption [158], uncontrolled inflammatory response [98,159], increased glutamate signaling [160] and, more recently, gut dysbiosis [161,162] in the development antidepressant therapy failure. Remarkably, some studies have suggested that hypercortisolism is an important factor associated with treatment-resistant unipolar depression. Furthermore, the highly resistant depressed patients showed an impaired response of the HPA axis to prednisolone and mineralocorticoid receptor antagonist, suggesting that there was a malfunctioning of the GRs and MRs [163,164]. Importantly, the increase in bacterial translocation, attested by measurement of 16S rRNA subunits of intestinal microbiota in the blood plasma, and in the immune response to LPS, including a rise in circulating pro-inflammatory cytokine levels and up-regulation of TLR4 signaling in peripheral mononuclear blood cells, was observed in patients with MDD [165,166,167]. Altogether, this evidence strongly suggests that both HPA axis hyperactivity and gut dysbiosis play an important role in the development of treatment-resistant depression observed in diabetic patients who have developed depressive disorders.

## 5. Conclusions

We postulate that hypercortisolism acts as a second hit in the onset of comorbid depression observed in diabetic patients, through an induction of neuroinflammation and an increase in glutamatergic transmission in the hippocampus. Furthermore, hypercortisolism in diabetes may be important to the development of dysbiosis and the break-down of the epithelial–intestinal barrier with consequent translocation of pathogenic bacterial products, such as LPS, into the bloodstream. Possibly, the activation of TLR4 in the hippocampus of diabetics induces an exacerbation in the neuroinflammation and glutamatergic transmission, resulting in the aggravation of depressive symptoms and in the development of the treatment-resistant depression (Figure 2). In this respect, new therapeutic strategies founded on the normalization of HPA axis activity and/or gut microbiota seem to be potentially practical approaches for adjuvant treatment of comorbid depressive disorder in diabetic patients.

## Figures and Tables

**Figure 1 biomolecules-13-01504-f001:**
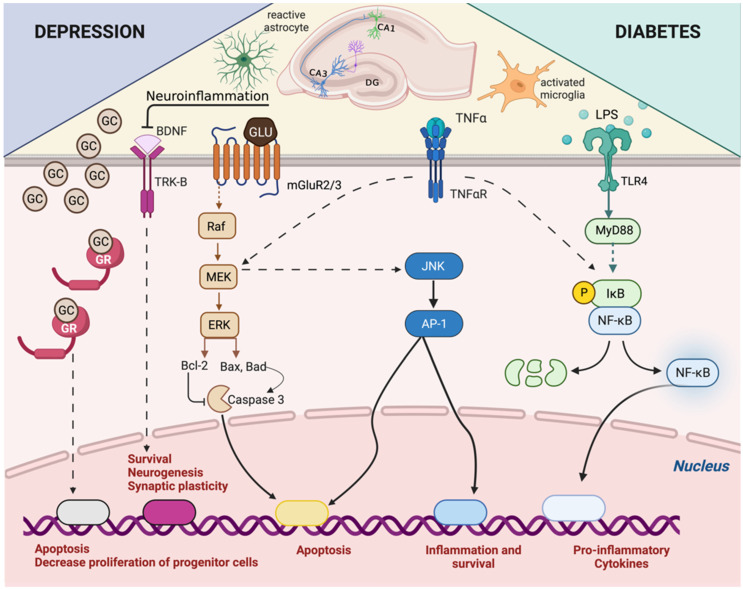
Scheme illustrating the main hippocampal mechanisms observed in the diabetes-related depression. The hippocampus represents a site of convergence for cellular and molecular changes observed in diabetes-associated depression. At the cellular level, there is an increase in the reactivity of astrocytes and microglia, a reduction in hippocampal neurogenesis, apoptosis of pyramidal neurons, and synaptic plasticity with dendritic retraction. At the molecular level, there is an increase in glucocorticoid signaling, a reduction in BDNF production, an increase in mGluR2/3 activity leading to caspase 3 activation and an increase in the TLR4/MyD88/NF-κB signaling pathway, with an augmentation in reactive oxygen species and transcription of pro-inflammatory cytokines, such as TNF-α, production. Together these pathways regulate gene transcription, increasing inflammation and apoptosis and decreasing progenitor proliferation, which in turn promotes a decrease in the hippocampal size.

**Figure 2 biomolecules-13-01504-f002:**
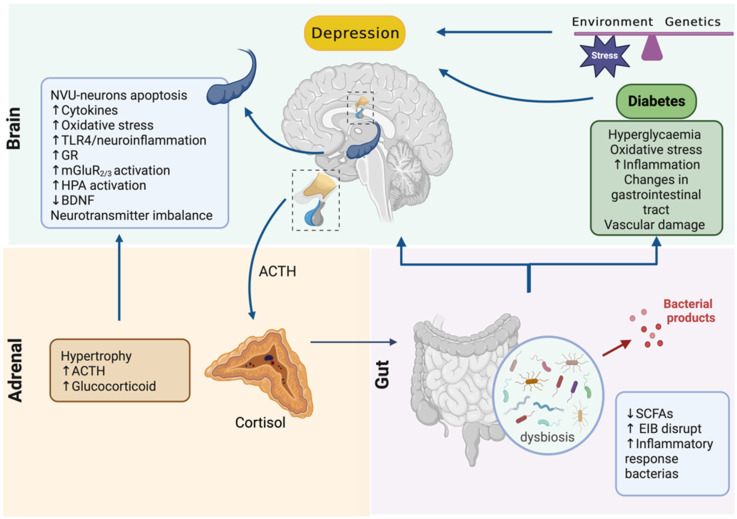
Adrenal–gut–brain axis imbalance contributes to aggravation of diabetes-related depression. In diabetes, hyperglycemia and hypercortisolism are associated with gut microbiota dysbiosis, as well as with changes in the central nervous system. Furthermore, the dysbiosis favors disruption of the endothelial intestinal barrier and the translocation of bacterial products into the blood, activating inflammatory signaling pathways. In the central nervous system, alterations in the physiology of the hippocampus are observed, associated with an imbalance in the neurotransmitters and a decrease in BDNF levels. Together, the exacerbation of neuroinflammation and glutamatergic transmission in the brain results in the worsening of depressive symptoms and the development of the treatment-resistant depression in diabetic patients.

## Data Availability

Not applicable.

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
