# Peer review of "The Role of the Adrenal–Gut–Brain Axis on Comorbid Depressive Disorder Development in Diabetes"

_biomolecules, 2023, doi:10.3390/biom13101504_

Round 1
Reviewer 1 Report
The present review discusses about the involvement of the adrenal-gut-brain axis (including microbiota) in diabetic patients suffering from depression. Authors propose hypercortisolism as the second hit involved in the susceptibility of diabetic patients to develop this mood disorder. Moreover, they postulate that the hyperactivity of HPA axis may explain the higher severity and resistance to the treatments of diabetics as well.
The manuscript is well-writen and the topic is relevant. In my opinion it could be improved discussing more in-depth some of the aspects addressed along the text. There are also concerns and questions to be solved:
Line 108: “possibly a breakdown of the intestinal epithelial barrier will be observed in diabetics”. Are there any studies showing direct or indirect measures of gut permeability in diabetic patients? If so, this information must be included in the manuscript.
Authors disscuss about the impact of a decrease in serum 5-HT levels and higher cortisol levels in patients with post-traumatic stress disorder (PTSD) (lines 148-150). In humans, most of serotonin (80-90%) is produced by enterochromaffin cells with the cooperation of gut microbiota. Do PTSD patients develop dysbiosis? More specifically, is there any study correlating the drop in 5-HT with dysbiosis in these patients? If so, this information must be included in the manuscript.
Adrenal exhaustion is associated with a decrease of circulating cortisol, is there any information about the impact of this condition on adrenal-gut-microbiota-brain axis?
Line 167: It appears “such as Clostridium, a butyrate producer bacteria”. On the one hand, Clostridium is a genera, which means that include many species. Thus, it is not “a“ single bacteria. On the other hand, do every bacteria from the genera Clostridium produce butyrate? If not, this point must be clarified.
Line 211: “Hyperglicemia and low-grade inflammation, which are observed in diabetic patients, disrupt epithelial intestinal barrier and BBB, especially in hippocampus”. This sentence must be revised and/or rewritten.
Line 218:I find quite interesting that metformin have the same effect for depressive symptoms in both diabetic and non-diabetic patients. Do metformin interact with HPA axis or gut microbiota?
The figure 1 includes several enzymatic pathways that must be briefly commented of the figure legend (at least).
Moderate editing of English language is required, since there are some expressions that complicate the understanding of the manuscript. For instance: line 77 “related to both a failure impaired in the negative feedback”; lines 80-81 “and an impaired in the negative feedback”; line 91 “the blocked of proinflammatory pathway…”., lines 113-114, etc.
Reviewer 2 Report
Some review analyses studies have reported the clinic depressive status and diabetic conditions in human plays comorbidity, however, very little information of the effects of adrenal-gut-brain axis on psychiatric status and diabetic conditions was reported. This reviewed work reported the differences between adrenal and gut axis in patients with depression and diabetic status. However, I do have some suggestions.
Neurodegenerative and metabolic diseases constitute a major problem of public health that is associated with increased risk of mortality and poor quality of life. Living environmental status has been considered as a major problem that worsens the prognosis of patients suffering from metabolic and neurodegenerative diseases. In this aspect, the present literature review work was aimed to critically collect and summarize all the available existing epidemiological and experimental data as far as concern the clinical impact of physiological assessment in psychiatric diseases, highlighting on the crucial role of diabetic status in psychiatric disease progression and management. According to the currently available epidemiological and clinical data, the mental status, especially in depression, of subjects seems to be affected by one direction of adrenal in the psychiatrics. However, the possible causes and discussion of gut microbiome and their other metabolites (i.e., indoles and secondary bile acids etc.) on T2DM status did not provide. Such information should be updated and discussed.
Overall, this text was a fair-organized review article in comorbidity of psychiatric and metabolic diseases, especially in depression and diabetes. The possible mechanism of the effects of adrenal hormones on gut microbiome and their metabolites on T2DM status were not mentioned and should be concluded and highlighted in the section of Discussion.
Round 2
Reviewer 2 Report
Authors responded the considerations appropriately. I have no further questions.